# Mechanistic Insights into the Biological Effects of Engineered Nanomaterials: A Focus on Gold Nanoparticles

**DOI:** 10.3390/ijms24044109

**Published:** 2023-02-18

**Authors:** Nhung H. A. Nguyen, Priscila Falagan-Lotsch

**Affiliations:** 1Institute for Nanomaterials, Advanced Technologies and Innovation, Technical University of Liberec (TUL), Studentsk. 2, 46117 Liberec, Czech Republic; 2Department of Biological Sciences, College of Sciences and Mathematics, Auburn University, Auburn, AL 36849, USA

**Keywords:** gold nanoparticles, molecular effects, nano–bio interactions, nanomedicine, nanotoxicology, omics

## Abstract

Nanotechnology has great potential to significantly advance the biomedical field for the benefit of human health. However, the limited understanding of nano–bio interactions leading to unknowns about the potential adverse health effects of engineered nanomaterials and to the poor efficacy of nanomedicines has hindered their use and commercialization. This is well evidenced considering gold nanoparticles, one of the most promising nanomaterials for biomedical applications. Thus, a fundamental understanding of nano–bio interactions is of interest to nanotoxicology and nanomedicine, enabling the development of safe-by-design nanomaterials and improving the efficacy of nanomedicines. In this review, we introduce the advanced approaches currently applied in nano–bio interaction studies—omics and systems toxicology—to provide insights into the biological effects of nanomaterials at the molecular level. We highlight the use of omics and systems toxicology studies focusing on the assessment of the mechanisms underlying the in vitro biological responses to gold nanoparticles. First, the great potential of gold-based nanoplatforms to improve healthcare along with the main challenges for their clinical translation are presented. We then discuss the current limitations in the translation of omics data to support risk assessment of engineered nanomaterials.

## 1. Introduction

First introduced in 1959 by the Nobel Prize Laureate physicist Richard Feynman, the concept of nanotechnology has been considered by many as the most significant technological breakthrough of the 21st century. Engineered nanomaterials (ENMs), defined by the International Organization for Standardization (ISO) as materials with at least one dimension between 1 and 100 nm [1], are at the cutting-edge of nanotechnology [2,3]. Due to their unique physicochemical properties and manifold variation possibilities, a myriad of ENMs has been developed and applied in virtually every industry and public endeavor, including electronics, automotive, textiles, renewable energy, environmental protection, and healthcare, bringing significant economic growth and innovation to our society [4,5]. The success of engineered nanoscale vaccines against COVID-19 is the most recent example of the transformative impact of nanotechnology on our lives [6]. Indeed, nanotechnology holds promise to revolutionize the biomedical field with limits only known to the imagination, providing new and very broad platforms for (or to improve) diagnosis, therapeutics, theranostics (combining diagnosis by imaging and therapeutics), repair, and regenerative medicine [7]. More than 25 years have passed since the first FDA-approved nanoplatform for drug delivery became available in 1995. This drug, a PEGylated liposomal doxorubicin (Doxil) used to treat different cancer types, is considered a milestone toward “the era of nanomedicine” [8]. Since then, the number of preclinical studies and clinical trials focused on nanomedicines for cancer and other human conditions has grown exponentially [7,9]. In addition, nanotechnology has a massive presence in our modern life. The advantages brought by materials in nanoscale make the use of ENMs extremely common in clothing, sunscreens, cosmetics, sporting equipment, batteries, food packaging, and dietary supplements, just to list a few [10].

However, in parallel to the societal benefits, the widespread use of ENMs along with the exponential growth in the global nanotechnology market, which is expected to reach 2314.81 kilotons by 2028 in terms of volume [11], has raised concerns regarding ENMs’ safety and impact on living systems. The properties that make ENMs technologically interesting—nanoscale size and high reactivity, among other unique characteristics—are the same that raise flags concerning their potential adverse effects on human health and the environment. The interactions of nanoparticles with biological systems are detrimental to the safety and efficacy of nanomaterials/nanomedicines [12]. Therefore, the limited understanding of the nano–bio interactions may hamper the use and commercialization of ENMs, impacting the development of the whole field of nanotechnology. Safety is essential for the approval of nanoplatforms by regulatory agencies and represents a major concern for their clinical translation [13]. Safety concerns are particularly evident considering inorganic nanoparticles, which can accumulate inside the body (low degradation/excretion), increasing the chances of long-term effects, still largely unknown. Moreover, the low efficacy of nanomedicines observed in clinical trials, another obstacle to their successful clinical translation, reflects our limited comprehension of the biological effects of ENMs [13,14,15]. Thus, a fundamental understanding of the interaction of ENMs with biological systems is of interest to nanotoxicology and nanomedicine. Moreover, it is of paramount importance in promoting the safe and sustainable development of nanotechnology.

The novel and variable properties of ENMs pose unique challenges in predicting their impact on living systems [16]. The biological effects and potential toxicity of ENMs can be significantly influenced by ENMs’ intrinsic factors related to their physicochemical properties, such as size, shape, chemical composition, surface, aggregation state, etc., as well as extrinsic factors, such as dose, cell type, and biological microenvironment, etc., (Figure 1) [17]. Hence, the use of conventional endpoint-based toxicity tests for macroscale chemicals, including cytotoxicity (cell death/cell viability) and DNA damage analysis, does not capture the complexities of the biological responses toward nanoscale materials. This implies the need for new methods to evaluate nano–bio interactions [18,19].

### Mechanistic Insights into the Biological Effects of ENMs—The Omics Era

In recent years, advanced approaches have been proposed to provide insights into the mechanisms underlying the biological responses to ENMs to ultimately support risk assessment [21,22]. These approaches include the use of omics techniques and systems toxicology, an integration of systems biology with toxicology, to describe molecular changes and perturbations in biological pathways and networks triggered by ENMs, which may lead to adverse effects. These predictive, mechanistic-based biological observations in toxicity evaluations for ENMs are aligned with the new paradigm for toxicity-testing proposed in 2007 by the National Research Council (US National Academy of Sciences) in the report “Toxicity Testing in the 21st Century: A Vision and a Strategy” [23,24]. The central part of this novel toxicology is to describe toxicity pathways, which can lead to a deep understanding of the molecular fundamentals and modes of action of chemicals and ENMs in humans and the environment.

In life sciences, the suffix “omics” means “a study of the totality of something”. Omics are comprehensive approaches for the analysis of complete molecular profiles of cells and tissues and include genomics, transcriptomics, proteomics, metabolomics, epigenomics, epitranscriptomics, etc. Omics allow studying how complex interactions between genes and molecules influence a phenotype, providing a comprehensive understanding of the biological system of interest. In the context of nanosafety, omics-based approaches have been used to investigate the mechanisms underlying the biological responses to ENMs at the molecular level, providing clues about the associations between ENM characteristics and biological features [25,26,27]. Omics data can also enable the development of biomarkers of exposure or the (early) effect of ENMs (before obvious phenotypic changes) [22,28].

Systems toxicology focuses on the understanding of complex interactions within biological systems and uses the “big data” derived from omics measurements combined with computational tools (modeling and bioinformatics) to describe the resilience of biological systems to perturbation by toxicants [29,30]. Since adverse outcomes resulting from toxicants are often caused by perturbations in biological networks at the systems level, the fundamental approach in systems toxicology involves the integrative analysis of multiple omics layers, bringing unprecedented insights into toxicological mechanisms [28]. Additionally, the knowledge generated by nanotoxicology studies can be used to guide the development of effective nanomedicines and boost their clinical translation.

In this review, we provide an overview of the literature available that best illustrates the application of omics and systems toxicology approaches in studies aiming to understand the biological effects of ENMs at the molecular level, focusing on gold nanoparticles. Since the number of reports on in vitro omics is much higher than that of in vivo experiments, only studies applying human and mouse cells as model systems for assessing the molecular changes and biological pathways perturbed by nanoparticles were considered. The great potential of gold nanoparticles to improve diagnosis and therapeutics, the clinical trials involving gold-based nanoplatforms, as well as the main challenges for their clinical translation are presented. Then, the current limitations to translate the omics data to support risk assessment of ENMs are discussed.

## 2. The Golden Age of (Nano) Medicine

Among the nanoparticles of interest for nanomedical applications, gold nanoparticles (AuNPs) are some of the most investigated. The reason behind such great interest is mostly linked to their fascinating tunable physicochemical properties resulting from surface plasmon resonance (SRP) [31,32,33], defined as the collective oscillations of conduction-band electrons on the surface of metals in nanoscale (Figure 2). These properties make AuNPs very attractive for a variety of biological applications, such as bioimaging, nano-sensing, molecular diagnostics, drug/gene delivery, photon-induced therapeutics, and theranostics [7,33]. Furthermore, the well-controlled synthesis procedures, the broad variety of shapes (spherical, rods, stars, shell, cages, etc.), and surface functionalization chemistries (polymers, peptides, nucleic acids, drugs, etc.) along with the fact that gold is commonly considered biologically inert also contribute to the popularity of AuNPs for applications in the biomedical field [34,35] (Figure 3).

The use of gold in medicine is not recent; historically, colloidal gold was used for medical purposes in the fifth and fourth centuries B.C.; in the Middle Ages, physicians and alchemists knew about the colloidal gold’s healing properties [37]. In the late 1920’s, gold salts were proposed to treat rheumatoid arthritis, a therapeutic approach used for about 70 years [38] until new and less toxic treatments were developed. Currently, gold nanoplatforms are being explored in preclinical studies, in vitro and in vivo, for a variety of medical applications, ranging from vaccine development to cancer therapy [39,40] (additional information about the biomedical applications of AuNPs can be found in refs. [41,42]). However, no gold-based nanoplatforms have been clinically approved by the FDA (US Food and Drug Agency) or EMA (European Medicines Agency), despite the optimism brought by the promising results in numerous preclinical studies. To date, only six gold-based nanoformulations have been under clinical trial investigation (https://clinicaltrials.gov/ (accessed on 1 December 2022) (Table 1), which is very low compared to the number of preclinical studies published every year [43].

Aurimune, a 7 nm PEGylated (poly(ethylene glycol)) colloidal gold particle loaded with TNF-α (tumor necrose factor alpha), was developed to treat patients with advanced and metastatic solid tumors, and has successfully completed Phase I (NCT00356980 and NCT00436410). AuroLase, gold-silica nanoshells coated with PEG (PEGylated AuroShell; ~150 nm diameter), was designed to maximally absorb near-infrared light and convert it to heat, and has been investigated for photothermal cancer therapy against lung (NCT01679470), head and neck (NCT00848042), and prostate tumors (NCT02680535) [44]. Despite the promising initial results for the treatment of prostate cancer [39], the AuroLase therapy pilot study in patients with primary and/or metastatic lung tumors was terminated for unknown reasons (https://clinicaltrials.gov/ (accessed on 1 December 2022). Another gold-based nanoplatform, NU-0129, is composed of spherical gold nanoparticles loaded with RNA interference (siRNA) specific for the oncogene Bcl2Like12 (Bcl2L12) and OEG (oligo(ethylene glycol)), and has been showing good potential for the systemic treatment of glioblastoma (NCT03020017). Besides cancer therapy, gold nanoplatforms have also been evaluated in clinical trials for applications in other human diseases or conditions. For instance, CNM-Au8, 13 nm gold nanocrystals in a drinkable bicarbonate solution, has been investigated as therapeutics for neurodegenerative diseases such as Parkinson’s disease (NCT03815916), relapsing multiple sclerosis (NCT03993171); and amyotrophic lateral sclerosis (NCT04098406). This formulation has showed significant positive results regarding the diseases’ progression [45]. Another nanoformulation, C19A3-GNP, comprises ultrasmall-AuNPs (size smaller than 5 nm) loaded with a proinsulin-derived peptide (C19A3), and has been investigated for immunotherapy delivery to treat type 1 diabetes (NCT02837094) [46]. Recently (2022), a new clinical trial involving AuNPs of size 8–28 nm in diameter with varying shapes, denominated as the Gold Factor, was initiated aiming to determine the clinical value of this gold-based nanoformulation for improving joint health, function, and quality of life for arthritis patients (NCT05347602).

Despite the exciting results observed in the early-phase clinical trials aforementioned, the approval of gold nanoformulations to the clinic is still a challenge.

### Main Challenges in Gold-Based Nanomedicines Clinical Translation

In drug development, including nanomedicines, the major causes of failure in clinical trials are related to the lack of effectiveness and poor safety profiles that were not predicted in preclinical studies. A survey of clinical translation of cancer nanomedicines pointed out that around 94% of new nanomedicines are successful in Phase I of the clinical trial, but the success rate drops to about 48% and 14% in Phases II and III, respectively. This phenomenon is associated with the poor/lack of efficacy and toxicity of the nanomedicines, which highlights our limited understanding of the behavior of ENMs in biological systems [47].

Regarding AuNPs, their life cycle in the organism has not been fully understood. Once inside the body, nanoparticles are covered by biomolecules (proteins, lipids, metabolites, etc.) that are adsorbed in their surface forming a corona that “hides” the surface ligand in targeted nanoplatforms. The composition of this biocorona is influenced by the composition of the biological matrices and by the physicochemical properties of ENMs, such as surface coating, size, shape, etc. [48,49]. It is well accepted that the biocorona plays important roles in the cellular uptake, biodistribution, and toxicity of ENMs in cells/organisms [50,51]. In vivo studies have shown that AuNPs can persist in the body for long periods of time with massive accumulation in the liver followed by the spleen, which may lead to unknown long-term toxic effects [52].

In addition to life cycle, little is known about the fate of AuNPs after their cellular internalization. Gold is widely considered “biologically inert” and then, supposed to be stable in biological environments. However, Balfourier et al. [53] recently evidenced in a long-term study (6 months) that AuNPs undergo a significant biotransformation (degradation followed by recrystallization) into the lysosomes of human fibroblasts few weeks after cells are exposed. The structures formed by the biotransformation of AuNPs are very similar to aurosomes, deposits of gold salts found in lysosomes of rheumatoid arthritis patients’ cells treated with gold. Based on these findings, the authors concluded that there is a shared metabolism of degradation between gold salts and AuNPs [53]. These observations pave the way for a better understanding of AuNPs’ life cycle in organisms, but also raise questions about the long-term fate of the aurosome-like structures, such as the availability of these structures to other cell types or organs, elimination pathways, and toxicity.

Moreover, toxicity is a subject of major concern for the clinical translation of gold, as well as other inorganic nanoparticles. Undoubtedly, the path to clinical implementation is more difficult for inorganic nanoplatforms due to their complex formulations and uncertainties with long-term side effects as well [54,55]. The investigations conducted regarding AuNPs’ toxicity generate contradictory data. For instance, a general trend for AuNPs toxicity is that small-sized nanoparticles (<5 nm diameter) present higher toxicity compared to larger ones due to their increased surface area and, consequently, augmented reactivity [56,57]; however, other studies did not confirm this trend [58,59]. Some studies have also highlighted the great importance of the surface chemistry characteristics, not size, in the induction of biological responses and toxicity triggered by AuNPs [60,61]. Clearly, there are still many gaps in the knowledge of the potential toxicity AuNPs may induce. Most available data are based on the acute responses to AuNPs using endpoint-based conventional cytotoxicity assays developed for bulk materials. It is known that chemicals and ENMs can impact multiple biological functions and cellular pathways without eliciting acute toxicological responses (sub-cytotoxic doses) [62,63]. This emphasizes the need to elucidate the molecular mechanisms involved in the responses to AuNPs with different physicochemical properties (different sizes, shapes, surface chemistries, etc.) to promote their effective clinical translation.

## 3. Insights into the Molecular Effects of AuNPs in Biological Systems

Here, we present an overview of the studies exploring the mechanisms underlying the biological responses to AuNPs based on omics changes (transcriptomics, proteomics, metabolomics, epigenomics, and epitranscriptomics) using human and mouse cells.

### 3.1. Transcriptomics

Transcriptomics allows the analysis of the changes in the expression of thousands of genes in a cell or organism through the detection of mRNA levels. Transcriptomics has been extensively applied to describe the effects and mechanisms of a potential toxicant [64,65]. Microarrays and next-generation sequencing (NGS), specifically RNA-seq, are the most established methods for profiling transcriptional changes.

Transcriptomic changes in human cells (non-transformed human dermal fibroblast cells, HDF, and prostate cancer cells, PC3) were determined by microarray after short-term exposure to different types of surface-modified nanospheres [66]. Significant alterations in the expression levels of genes related to proliferation, angiogenesis, and metabolism were observed in HDF cells exposed to a low dose of AuNPs (0.1 nM), while genes implicated in inflammation, angiogenesis, proliferation, apoptosis regulation, survival, and invasion were dysregulated in PC3 cells exposed to 1 nM of the same AuNPs. These changes in gene expression were induced in a surface chemistry-dependent manner, suggesting the great complexity that involves surface coating and nano–bio interactions. Pronounced changes in the expression levels of genes related to metal ion binding, antioxidant pathways (upregulated), and selenium homeostasis (downregulated) were detected by microarray after the exposure of Caco2 cells to a cytotoxic dose (300 μM) of spherical 5 nm-AuNPs for 72 h [67]. These results describe the possible mechanism of cytotoxicity induced by the AuNPs in cancer cells, and might be used to explore their potential anti-cancer properties. The exposure of MRC-5 cells to sub-cytotoxic doses of AuNPs (rods) induced a concentration-dependent attenuation in the cell growth detected by a label-free, real-time cell-monitoring platform that measures electrical impedance. A transcriptomic analysis of these cells exposed to AuNPs (360 ng/mL) for 24 h showed the upregulation of genes involved in DNA damage response, cell cycle regulation, and antioxidant pathways, providing insights into the mechanisms that might be contributing to the attenuation of MRC-5 cell proliferation and resistance to cytotoxicity [68]. Transcriptomics analysis performed using microarray at different time points (1 d, 2 wk, and 2 mo) after the exposure of primary human skin fibroblasts to 4 nm AuNPs revealed a long-lasting oxidative response linked to the biotransformation of AuNPs reflected by the enrichment of genes related to oxidative stress (antioxidant pathways), exogenous stress, and immune response in all time points. Interestingly, most genes upregulated at the later time points (2 wk and 2 mo) were not overexpressed on Day 1, suggesting that short-term studies do not fully capture the specific biological responses to AuNPs [53].

The increased amount of data regarding the transcriptional-based mechanism of action of ENMs (tMOA) prompted the development of INSIdE NANO, a computational tool that provides a systemic biology framework to contextualize the tMOA signatures of ENMs with respect to human diseases, drug treatments, and chemical exposures [69]. By using this novel approach, the authors highlighted intriguing tMOA similarity patterns between AuNPs and neurodegenerative disorders such as Parkinson’s disease and amyotrophic lateral sclerosis (ALS). It is well known that AuNPs may induce oxidative stress in vitro and in vivo in brain cells and, in some cases, a concomitant antioxidant response is not observed, increasing the chances of cell damage and diseases. Moreover, neurotoxicity has been reported in patients with rheumatoid arthritis (RA) who received gold treatment [70,71]. However, it is still being determined whether gold in the nanoscale may elicit similar effects. Transcriptomic changes induced by AuNPs in cells are shown in Table 2.

Transcriptomics data have also been used to identify the biomarkers of ENMs’ exposure and effect. Gene signatures for AuNPs exposure were explored by microarray analysis of several human cell lines representing the primary routes of ENMs’ exposure such as inhalation (A549 cells) and ingestion (HEK293, HepG2, and AGS cells) [72]. After a 24 h exposure to a sub-cytotoxic dose of AuNPs (360 ng/mL) with different diameters—39, 41, and 45 nm—the TSC22D3 (related to inflammation, apoptosis, and cell cycle progression), TRIB3 (cell stress response), PCK2 (modulation of cell survival during stress), and DDIT4 (DNA damage response) genes were significantly upregulated in all cells tested. The same was not observed when the same cell lines were challenged with an equal concentration of micro-Au and nano-TiO_2_, suggesting that the four gene signatures could be potentially used as biomarkers for early AuNPs exposure [72]. Biomarkers can be used to facilitate the development of new tools for biomonitoring toxicants (ENMs) exposure/effects and might be integrated into the risk assessment [73,74]. Thus, the identification of molecular signatures reflecting the ENMs effects and exposure, particularly in the long-term using realistic low-dose exposures, is urged.

### 3.2. Proteomics

The proteome is the complete set of proteins produced by a cell, tissue, or organism. Proteome-wide studies provide a close link to cell phenotypes under a particular environmental condition. However, proteomics approaches have been mainly applied in nano–bio interaction studies to qualitatively characterize the layer of proteins adsorbed to the surface ENMs, the protein corona, which is critical for ENMs’ uptake by cells, intracellular localization, and toxicity [75,76,77]. Quantitative proteomics-based approaches, particularly mass spectroscopy (MS), are beginning to be explored as tools for mechanistic studies of biological responses to ENMS [78,79]. Proteomic analysis of human dermal fibroblasts-fetal (HDF-f) treated with 20 nm citrate AuNPs was performed using two-dimensional difference gel electrophoresis (2D-DIGE) and mass spectroscopy (MS). Pathway analysis showed that the most differentially expressed proteins were related to signal transduction, cytoskeleton, energy metabolism, and oxidative stress cell transcription factor, among others [80]. Using the same proteomic approaches along with protein microarrays to detect changes in the protein phosphorylation profiles (phosphoproteome), Tsai et al. reported that proteins involved in the endoplasmic reticulum (ER) stress response were the most dysregulated by the exposure of K562 cells to a low dose of “naked” AuNPs (no surface coating) [81]. Moreover, these AuNPs induced an increase in the phosphorylation levels of proteins in receptor tyrosine kinase pathways (RTK activation) associated with the ER. The unmanageable ER stress triggered by AuNPs resulted in cell death, demonstrating a potential use of these AuNPs in nanomedicines for cancer therapy. The quantitative proteomic profiling of MRC-5 lung fibroblasts co-cultured with small airway epithelial cells (SAECs) pretreated with 1 nM of citrate-coated AuNPs revealed a dysregulation of proteins mainly related to cell adhesion and extracellular matrix (ECM)/cytoskeleton remodeling that may affect lung function [82]. To study the effects of bimodal AuNPs coated with amine groups (NH2) on the innate immune system, Tarasova et al. employed liquid chromatography–tandem mass spectrometry (LC-MS/MS) to profile protein expression patterns in THP-1 cells after a short exposure to AuNPs [83]. The cell treatment triggered the upregulation and activation of NF-κB, a key inflammatory mediator. Furthermore, TIPE2, an NF-κB inhibitor, was downregulated and identified as a direct target of AuNPs, pointing toward direct protein binding as a possible mechanism underlying the proinflammatory effects of AuNPs [83]. The proteomics changes induced by AuNPs in cells are summarized in Table 3.

### 3.3. Metabolomics

Metabolites are typically defined as small molecules (mass < 1500 Da) that are required for metabolic reactions that drive cellular/organism growth and functions [84,85] Metabolomics involves the profiling of metabolites in biofluids, cells, and tissues and, because metabolites are extremely sensitive to subtle alterations in biological pathways, metabolomics holds great promise for understanding the molecular mechanisms underlying the biological responses to ENMs [86,87]. Two analytical approaches are used for metabolomics studies: untargeted and targeted. While untargeted metabolomics is focused on the global metabolomic profile aiming to identify and quantify the broadest range of metabolites possible in a biological sample, providing the possibility of discovering novel biomarkers and molecular key events in adverse outcome pathways (AOP), the targeted approach focuses on a limited, well-defined set of metabolites, providing a deeper understanding of the selected metabolites [88]. A variety of separation methods —liquid chromatography (LC), gas chromatography (GC), capillary electrophoresis (CE), mass spectrometry (MS), and nuclear magnetic resonance (NMR)— are currently employed for metabolomics analysis. A metabolomics study using both GC-MS and LC-MS/MS techniques reported the depletion of intracellular metabolites in HepG2 cell line exposed to AuNPs, presenting different surface chemistries (citrate, poly-(sodium styrene sulfonate)-PSSNa, and poly-vinylpyrrolidone—PVP) [89]. The decrease in metabolite concentration was observed regardless of AuNPs’ surface coating, suggesting that AuNPs bind certain metabolites with or without displacing the surface ligand. The impact of spherical AuNPs on the metabolome was also investigated in HDF cells [90]. By employing the MetaboAnalyst online tool (https://www.metaboanalyst.ca)the metabolic pathways affected by the exposure of HDF cells to AuNPs were mainly related to the glutathione metabolism. Indeed, glutathione, which plays an important role in protecting cells from an oxidative environment, was identified as the key metabolite induced by AuNPs, reflecting a protective response in HDFs to reduce AuNPs-mediated oxidative damage [90]. Interestingly, when the GC-MS-based metabolomics approach was used to compare the toxicological effects of AuNPs with different shapes, spheres, and stars (of similar ~40 nm diameter and coated with 11-mercaptoundecanoic acid) 24 h after an intravenous administration of a single dose (1.33 × 10^11^ AuNPs/kg) to rats, an increase in the glutathione levels was also observed in the liver, where both types of AuNPs were preferentially accumulated [91]. The metabolic profile of nanospheres versus nanostars was clearly discriminated, particularly by the differences in metabolic pathways related to the metabolism of fatty acids, pyrimidine, purine, arachidonic acid, biotin, glycine, and the synthesis of amino acids [91]. Xu et al. [92] also reported alterations in the glycine, serine, and threonine metabolism pathways after the 24-h-treatment of TM-4 mouse Sertoli cells with AuNPs (rods, 10 nm width, 40 nm length), corroborating the idea of a general panel of biological pathways associated with AuNPs toxicity [92]. Recently, untargeted metabolomics experiments revealed that 80 nm double-stranded DNA-modified AuNPs (AuNPs-dsDNA) can possibly regulate the metabolic reprogramming of PC3 and DU145 prostate cancer cell lines mainly through the lipid metabolic pathways [93]. Since metabolic reprogramming plays an important role in the development of prostate cancer, this result provides an important basis for future research on the characteristic targeted design of nanomaterials for cancer metabolism [93]. The metabolomics changes caused by cells exposure to AuNPs are shown in Table 4.

### 3.4. Epigenomics

The epigenome covers heritable changes in gene expression without direct alteration in the DNA sequence. DNA methylation, histone modifications, and noncoding RNAs are included in the panel of epigenetic mechanisms. The epigenome is deeply responsive to the environment and may be altered by a myriad of factors, such as stress, chemicals, etc. Since substantial evidence suggests that epigenetic mechanisms have a critical role in determining adverse health outcomes, regulatory agencies, such as EPA, have been discussing the incorporation of epigenetic data to identify chemicals with the potential to cause adverse effects and support human health risk assessment [94,95,96]. Several in vivo and in vitro studies demonstrate the disruption of the epigenome as a result of exposure to ENMs [97,98,99], but studies focusing on AuNPs are still very limited (Table 5).

#### 3.4.1. DNA Methylation and Histone Modifications

DNA methylation is one of the main epigenetic mechanisms for gene regulation and represents the most investigated epigenetic parameter in studies focused on the impact of ENMs in biological systems. DNA methylation is a post-replication modification that predominantly involves the covalent addition of a methyl (CH3) group, almost exclusively found in the 5′ positions of cytosines (5mC), present in the dinucleotide sequence CpG in mammal cells. The methylation of cytosines within CpG sequences, and their subsequent interaction with methyl-CpG-binding proteins (MBDs), may induce chromatin conformational modifications and inhibit the access of the transcriptional machinery to gene promoter regions, thus altering gene expression levels [100]. In the genome, CpG dinucleotides are generally methylated, and as a rule, methylation in promoters suppresses gene transcription [100,101]. The perturbations in DNA methylation patterns and in DNA methylation enzymatic machinery are associated with many human diseases such as cancer and neurological disorders [102,103]. Human embryonic stem cells (hESCs) exposed to thiolate-capped 4 nm AuNPs presented a dramatic decrease in global DNA methylation (5mC) after 24 h of exposure [104]. The global decline of DNA demethylation levels (removal of 5mC) was observed when MGC-803 and HEK293FT cells were treated with a non-cytotoxic dose of glutathione-based AuNPs, leading to changes in the expression of genes involved in cell adhesion, migration, proliferation, differentiation, and cell apoptosis [105]. This change in the epigenome was caused partially by ROS activation and oxidative stress generated by the AuNPs exposure. Citrate-capped AuNPs with different sizes (5, 60, and 250 nm spheres) at two different doses (5 and 50 µg) did not elicit changes in global DNA methylation patterns in vivo (mouse lungs). However, at the highest dose, the 60 nm AuNPs induced changes in the methylation profile (hypo- or hypermethylation) of specific gene promoters in mouse lung tissue after 48 h [106].

Histone modifications are posttranslational covalent modifications of histone protein tails also involved in the epigenetic regulatory mechanism since these chemical modifications determine the interaction between histones and other proteins and regulate the chromatin structure as well [107]. Modifications in histones include phosphorylation, methylation, acetylation, ubiquitylation, and sumoylation. Like DNA methylation, histone modifications are highly dynamic processes [108,109]. To date, acetylation, phosphorylation, and methylation are the most studied histone modifications. The exposure of MRC-5 cells to 20 nm citrate-capped AuNPs for 72 h decreased histone H3 lysine 27 trimethylation (H3K27me3) associated with transcriptional repression [110]. Using mass spectroscopy (LC-MS) followed by chromatin immunoprecipitation sequencing (ChIP-seq), a recent study also reported the effect of AuNPs in the H3K27 methylation profile in a shape-dependent way, with star-shaped “spiky” AuNPs, but not the spherical ones, inducing demethylation of di- and tri-methylation lysine 27 (H3K27me2/3) in dTPH-1 and A549 cell lines [111]. These results suggest that the architecture of nanoparticles plays an important role in reprogramming the epigenome and may be explored to guide the development of nano-based therapies to target aberrant epigenetic patterns associated with diseases.

#### 3.4.2. Noncoding RNAs—miRNAs

MicroRNAs (miRNAs), the most studied family of noncoding RNAs, are short (18–25 nucleotides), single-stranded molecules that mediate the regulation of gene expression by inducing translation repression and/or degradation of mRNA targets [112]. The human genome encodes approximately 2600 miRNAs, and it is estimated that up to 60% of protein-coding genes are modulated by miRNAs at the transcriptional level [113]. Since miRNAs are involved in the regulation of virtually all biological pathways, from the development to the maintenance of homeostasis, miRNA dysregulation has been related to pathogenesis [114,115,116]. miRNAs also play a fundamental role in cellular responses to environmental chemicals [117]. Despite the importance of miRNAs in the (nano) toxicology research and the fact that changes in miRNA expression profiles have been reported after exposure to ENMs [118,119,120,121,122], few studies have addressed the impact of AuNPs on miRNA dysregulation (Table 5).

Microarray analysis of miRNA isolated from the blood of rats at 1 week and 2 months after 20 nm AuNPs injection revealed short- and long-term changes in miRNA expression levels [123]. Transplacental epigenetic effects (miRNA expression changes) were observed in mouse fetal tissues after pregnant mice were treated with 100 nm citrate-capped AuNPs but not with 40 nm ones, highlighting the size-dependent effects of AuNPs in the epigenome [124]. A non-cytotoxic dose of AuNPs (20 nm; citrate-capped) induced big changes in the miRNA expression profile in non-transformed HDF (human dermal fibroblasts) cells in a short-term (1, 4, and 8 h) [124]. Biological pathway analysis revealed that the dysregulated miRNAs were mainly related to cellular metabolic processes, the mRNA-processing pathway, and the MAPK-signaling transduction pathway involved in the regulation of cell proliferation, differentiation, cell survival, and other critical cellular processes [124]. Using a network-based approach to evaluate changes in the miRNome detected by next-generation sequencing, Falagan-Lotsch and Murphy also reported dysregulation of co-expressed miRNAs mainly involved in signal transduction pathways in HDF cells 20 weeks after chronic and acute (non-chronic) treatments to four AuNPs (different shapes and surface chemistries) at a very low dose (0.1 nM) [63]. The acute (24 h) exposure to AuNPs was previously shown to induce ER stress in HDF cells in the long term [62], and led to significant perturbations in miRNA networks related to the modulation of cell proliferation, with cells under this exposure condition potentially suppressing proliferative signaling pathways through miRNAs [63]. It is known that ER stress is implicated in ROS production, which is involved in the activation of proliferative signaling pathways [125] but no increase in HDF proliferation rates was observed after the acute treatment with any AuNP. Therefore, the authors suggested that, in this case, the miRNA dysregulation is a cellular response to the stress provoked by acute exposure to AuNPs to restore cell hemostasis [63]. This study highlights the critical role of miRNAs in the long-term cellular responses to AuNPs, bringing new insights into the molecular changes underlying the biological responses to AuNPs in a more realistic scenario of low-dose exposure.

### 3.5. Epitranscriptomics

As in DNA, reversible chemical modifications are also known now to occur in RNAs. These posttranslational RNA modifications are referred to as epitranscriptomics modifications [126]. The epitranscriptome functions to regulate gene expression and play essential roles in RNA structure, processing, stability, and translation [127]. The methylation of adenosine at the N6 position (m6A) is the most common and the best characterized epitrancriptomics mark in eukaryotic mRNAs [128]. Changes in the m6A pattern, as well as in the m6A modification machinery have been associated to the stress responses to environmental factors and the development of diseases, including many types of cancer [129,130]. Very little is known about the impact of ENMs on the epitranscriptome (Table 5). A preclinical study showed that low doses of gold nanorods (AuNRs) with different aspect ratios functionalized with chitosan and a 12-mer peptide induced global m6A hypomethylation and posttranscriptional regulation (suppression) of genes related to glycolysis, hypoxia, and immune checkpoint pathways in leukemia cells [131]. Notably, the combined treatment with AuNRs and tyrosine kinases inhibitors (TKIs) eliminated the m6A-mediated TKIs cancer resistance in vivo, providing the proof-of-concept for the use of nanoparticles as an epigenetic drug for cancer therapy by targeting the epitranscriptome [131]. Recently, Pan et al. reported changes in the m6A pattern (hypomethylation) evaluated by m6A-sequencing (m6A-seq) in HEK293T cells mRNAs after exposure to bovine serum albumin (BSA)-templated nanoparticles, including small AuNPs (3 nm diameter), and these results can be partially explained by the abnormal aggregation of m6A-related enzymes induced by the exposure to the nanoparticles [132]. Interestingly, the genes presenting reduced m6A were mostly involved in TGF-beta signaling, critical for cell proliferation, differentiation, and apoptosis. Based on these findings, the authors demonstrated the potential of epitranscriptomics analysis for the toxicity evaluation of ENMs for clinical translation.

**Table 5 ijms-24-04109-t005:** Examples of epigenetic and epitranscriptomic changes induced by AuNPs in cells.

	Shape	Size	Surface Chemistry	Cell Type	Exposure	Method	Ref
**Epigenetics**							
DNA methylation							
	Spheres	4 nm	thiol	hESCs (human embryonic stem cells)	10 µg/mL; 24 h	Immunoprecipitation-based colorimetric assay	[104]
	Spheres	4–5 nm	L-Glutathione (L-GSH)	HEK293FT (human embryonic kidney 293) and MGC-803 (human gastric carcinoma)	100 µg/mL; 48 h	Dot blot assay	[105]
	Spheres	5, 60, and 250 nm	citrate	Lung tissue (BALB/c mouse)	5 and 50 µg; 48 h	LC-MS ^1^; bisulfite pyrosequencing	[106]
Histone modifications							
	Spheres	20 nm	citrate	MRC-5 (human normal lung fibroblasts)	1 nM; 72 h	Immunofluorescence	[110]
	Spheres, Stars	~20 nm, ~45 nm	Bis (*p*-sulfonatophenyl)phenylphosphine dehydrate dipotassium salt (BSPP)	dTHP-1 (differentiated human leukemia monocytic cell) and A549 (human lung adenocarcinoma)	1 × 10^11^ partilces/mL; 24 h	LC-MS ^1^ followed by chromatin immunoprecipitation sequencing (ChIP-seq)	[111]
miRNAs							
	Spheres	20 nm	citrate	Blood cells (rats)	1 wk; 2 mo	Microarray	[133]
	Spheres	40 nm; 100 nm	citrate	Swiss mice fetal liver and lungs	3.3 mg/kg (4 doses); 18 d of Swiss female mice gestation	Microarray	[123]
	Spheres	20 nm	citrate	HDF (human normal dermal fibroblast)	200 µM;.1, 4 and 8 h	RNA-Seq	[124]
	Spheres, Rods	20 nm; 16 nm width, 46 nm length	citrate, poly (acrylic acid) (PAA); poly (ethylene glycol) (PEG)	HDF (human normal dermal fibroblast)	0.1 nM; 24 h and 20 wk	RNA-Seq	[63]
**Epitranscriptomics**							
	Rods	21 nm width, 130 nm length	chitosan and 12-mer peptide	AML (Acute myeloid leukemia cells	0.25 nM; 6 h	m6A-Seq followed by gene-specific m6A-qPCR and LC-MS/MS ^2^	[131]
	Spheres	3 nm	bovine serum albumin (BSA)	HEK293T (human embryonic kidney 293)	200 μg/mL; 24 h	m6A-Seq followed by gene-specific m6A-qPCR	[132]

^1^ LC-MS: Liquid chromatography–mass spectrometry; ^2^ LC-MS/MS: liquid chromatography–tandem mass spectrometry.

### 3.6. Multi-Omics

An integrative analysis of multi-omics data is emerging to further elucidate the underlying molecular pathways affected by ENMs in living systems [134,135,136]. A combination of proteomics and metabolomics approaches was used to evaluate the effects of AuNPs, 5 and 30 nm citrate-capped spheres in Caco 2 cells after 72 h exposure. By integrating the multi-omics data with bioinformatic tools, biological pathways such as cell degeneration, cell morphology, and cell growth and proliferation were found dysregulated by the AuNPs treatment [137]. A multi-omics study using transcriptomics and proteomics data coupled with cell-based validation assays showed that the 24 h treatment with cationic ammonium-modified AuNPs, but not the anionic (COOH-modified) or neutral (PEGylated) ones, elicited mitochondrial dysfunction causing cell death with features of both necrosis and apoptosis in human monocytic THP-1 cells [138]. Moreover, the treatment triggered autophagy, a pathway involved in cell survival, suggesting a mechanism of cellular defense against the insult provoked by the cationic AuNPs. Based on the results, it is worth stressing the importance of intrinsic physicochemical properties, surface chemistry in this case, as determinants of the biological effects of NPs in living systems. The integrative multi-omics studies of the effects of AuNPs in cells are summarized in Table 6.

Based on the omics results presented in this review, the cellular exposure to AuNPs provokes molecular perturbations represented by changes in the profile of transcripts, proteins, metabolites, and epigenetics/epitranscriptomics markers. Coupled with bioinformatics analysis, omics studies identified multiple biological pathways perturbed by AuNPs mostly in sub-cytotoxic concentrations, anticipating effects not captured by conventional cellular assays. Overall, perturbations in pathways related to cell proliferation, adhesion, death (apoptosis), and metabolism were reported at multiple molecular levels, even when low doses of AuNPs were applied. The increased levels of ROS and the generation of an oxidative environment in cells seem to play an important role in the molecular changes and dysregulation of biological pathways induced by AuNPs. This is exemplified by the overexpression of antioxidant genes and proteins [53,67,68,80,81] and increased levels of glutathione [90,91], cellular attempts to restore homeostasis, and decrease in 5-hydroxymethylcytosine (5hmC), an epigenetic marker triggered by ROS [105]. While studies using conventional assays have reported conflicting results about the potential of AuNPs to cause oxidative stress, mechanistic approaches suggest that AuNPs lead to the dysregulation of redox-sensitive pathways and disturb cell homeostasis. Further functional studies are needed to validate these omics findings. Since most of the publications are focused on short-term exposures/effects, the long-term effects of the exposure to AuNPs and their potential to cause adverse biological outcomes are still largely unknown.

The reader can notice that citrate spheres were the preferred type of AuNP used in the in vitro omics studies. Citrate is considered a relatively “safe” surface chemistry to cells. However, citrate-capped AuNPs with different sizes evoked short- and long-term changes in the molecular profile of different cell types, leading to the disruption of cell homeostasis, even at a very low dose (0.1 nM) [63,110,124,137].

Novel and more specific biological responses induced by AuNPs, such as metabolic reprogramming and inflammatory effects, were mostly related to the intrinsic properties of the AuNPs tested, particularly shape and surface chemistry. These results emphasize the importance of “as-made”, as well as the biotransformed ENMs’ characterization in the understanding of nano–bio interactions.

## 4. Conclusions

Despite the enormous potential to improve healthcare, from the diagnosis to the treatment and prevention of diseases, the full benefits of nanotechnology in the biomedical field have not yet been achieved. The safety of ENMs has been a global concern [139], particularly addressing inorganic nanoparticles. Gaps in knowledge regarding the potentially harmful effects of ENMs have overshadowed the great benefits of their applications in nanomedicine.

Efforts have been made to fulfill the knowledge gaps for the safe adoption of nanotechnology. Understanding the impact of ENMs on biological networks and their potential adverse outcomes is required to predict the potential health risks associated with exposure to ENMs. Thus, omics approaches and systems toxicology have been progressively applied in nanotoxicology studies to potentially characterize the mechanism of action of ENMs, even at low doses, and identify adverse outcomes beyond phenotypes alteration. These approaches open new possibilities for the safety assessment of ENMs [22]. Currently, the use of omics technologies in nanotoxicity testing is still limited. These limitations are exemplified in the studies on AuNPs presented in this review. The reader may have noticed the lack of standardization of experimental/technological parameters: the use of different cell lines, different times of exposure, high doses of nanoparticles, and different omics methodologies. Reproducibility, integration and interpretation of (multi)-omics data in the context of biological function are also current challenges. To develop predictive models based on a system biology approach, multi-omics analyses coupled with appropriate bioinformatics tools are required. Thus, some expertise to interpret the big data along with computational resources is needed. Moreover, many studies investigate the effects of poorly characterized AuNPs or the effects of overall exposure without taking into consideration the role of the physicochemical parameters of AuNPs on the molecular profiles. Thus, well-designed experiments using well-characterized ENMs, relevant concentration ranges, and relevant cell types exposed under realistic exposure conditions are necessary to provide good quality and applicability of omics data to support risk assessment [140]. Addressing AuNPs specifically, as gold-based nanomaterials are slowly degraded, toxicological assessments must be conducted over extended periods of time. Furthermore, since the predictive value of omics results is not entirely clear, determining the functional significance of the omics data, as well as identifying molecular signatures induced by ENMs are important challenges. The adoption of a network view of the molecular effects of well-characterized ENMs is highly encouraged.

In summary, by integrating data from omics and systems toxicology with the physiochemical properties of ENMs, it is possible to contextualize ENMs’ mechanisms of action with respect to human diseases, enabling the development of safe-by-design nanomaterials, and boosting ENM-driven innovations. Additionally, the knowledge of the interactions of ENMs with biological systems at multiple levels of the molecular organization can be used to improve the efficacy of nanomedicines and open new avenues for nanotechnology applications in the biomedical field.

## Figures and Tables

**Figure 1 ijms-24-04109-f001:**
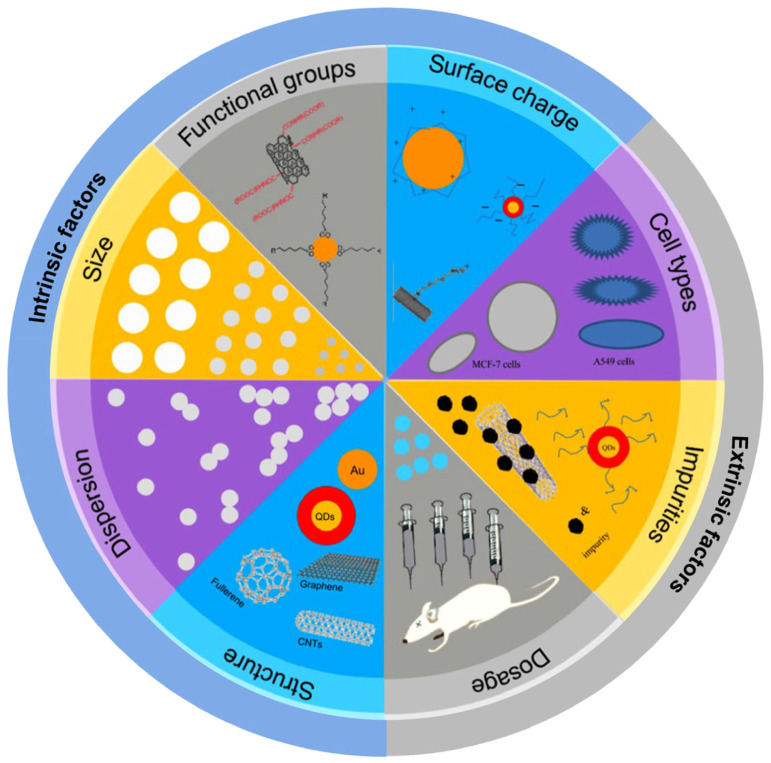
Intrinsic and extrinsic factors that can modulate the toxic effects of ENMs (reproduced and modified with permission from Zhao et al., ref. [20]. Copyright 2015, Elsevier).

**Figure 2 ijms-24-04109-f002:**
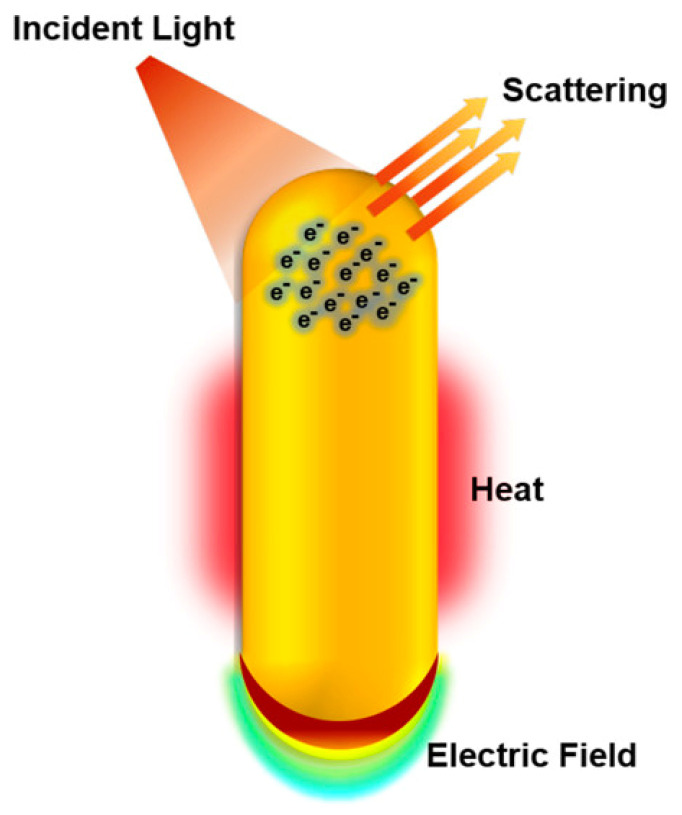
Physicochemical properties of gold nanoparticles explored in biological applications. Under illumination at or near the plasmon bands, gold nanoparticles absorb and scatter light, produce local electric fields, and can generate heat. These properties enable gold nanoparticles to be applied in bioimaging, surface-enhanced spectroscopies for sensing and molecular diagnostics, and photothermal therapy to destroy nearby cells. Due to their easy surface functionalization, gold nanoparticles are also explored for gene and drug delivery (reproduced with permission from Murphy et al., ref. [33]. Copyright 2019, American Chemical Society).

**Figure 3 ijms-24-04109-f003:**
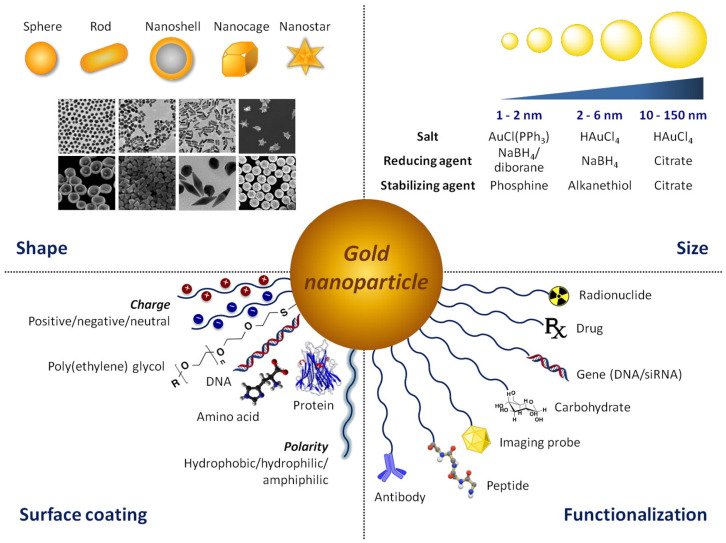
The versatility of gold nanoparticles. Gold nanoparticles present tunable sizes and shapes, and straightforward surface modification, enabling fine-tuning of their properties and functions. (reproduced with permission from Her et al., ref. [36]. Copyright 2017, Elsevier).

**Table 1 ijms-24-04109-t001:** Gold-based nanoplatforms investigated in clinal trials (https://clinicaltrials.gov/ (accessed on 1 December 2022).

Name	Composition	Investigated Application	Clinicaltrials.Gov ID (Phase)
Aurimune	7 nm PEGylated colloidal gold particles loaded with TNF-α	treatment of advanced and metastatic solid tumors	NCT00356980; NCT00436410 (both Phase I)
AuroLase	~150 nm diameter gold-silica nanoshells coated with PEG	photothermal therapy against lung, head and neck and prostate tumors	NCT01679470 (not applicable) NCT00848042 (not applicable) NCT02680535 (not applicable)
NU-0129	gold nanoparticles loaded with siRNA specific for the oncogene Bcl2L12, and OEG	treatment of glioblastoma	NCT03020017 (early Phase I)
CNM-Au8	13 nm gold nanocrystals	Parkinson’s disease, multiple sclerosis, amyotrophic lateral sclerosis	NCT03815916 (Phase II) NCT03993171 (Phase II) NCT04098406 (Phase II)
C19A3-GNP	ultrasmall-AuNPs (less than 5 nm) loaded with a proinsulin-derived peptide	treatment of type 1 diabetes	NCT02837094 (Phase I)
Gold Factor	8–28 nm AuNPs with varying shapes	treatment of arthritis	NCT05347602 (not applicable)

**Table 2 ijms-24-04109-t002:** Examples of transcriptomic changes induced by AuNPs in cells.

Shape	Size	Surface Chemistry	Cell Type	Exposure	Method	Ref
Spheres	20 nm	citrate, poly(allylamine hydrochloride) (PAH), and lipid coatings combined with alkanethiols or PAH	HDF (human dermal fibroblasts); PC3 (human prostate cancer)	0.1 nM (HDF); 1 nM (PC3); 24–48 h	Microarray	[66]
Spheres	5 nm	citrate	Caco2 (human epithelial colorectal adenocarcinoma)	300 μM; 72 h	Microarray	[67]
Rods	10–40 nm length	no data	MRC-5 (human normal lung fibroblasts)	360 ng/mL; 24 h	Microarray	[68]
Spheres	4 nm	citrate	HDF (primary human dermal fibroblasts)	no data; 1 d, 2 wk, 2 mo	Microarray	[53]
Spheres	39, 41, and 45 nm	no data	A549 (human lung adenocarcinoma); HEK293 (human embryonic kidney 293), HepG2 (human hepatocellular carcinoma), and AGS (human gastric adenocarcinoma)	360 ng/mL; 24 h	Microarray	[72]

**Table 3 ijms-24-04109-t003:** Examples of proteomics changes induced by AuNPs in cells.

Shape	Size	Surface Chemistry	Cell Type	Exposure	Method	Ref
Spheres	20 nm	citrate	HDF-f (human dermal fibroblasts-fetal)	200 μM; 1, 4 and 8 h	2D-DIGE/MS ^1^	[80]
Spheres	5.9 nm	no surface coating (“naked”)	K562 (human chronic myelogenous leukemia)	2.95 nM; 48 h	2D-DIGE/MS ^1^; protein microarray	[81]
Spheres	20 nm	citrate	MRC-5 (human normal lung fibroblast) in co-culture with SAECs (small airway epithelial cells)	1 nM, 72 h (SAECs)	LC-MS/MS ^2^	[82]
Spheres	10–20 nm and 32–54 nm (bimodal)	amine groups (NH_2_)	THP-1 (human leukemia monocytic cell)	15 μg/mL; 3 h	LC-MS/MS ^2^	[83]

^1^ 2D-DIGE/ MS: two-dimensional difference gel electrophoresis followed by mass spectroscopy. ^2^ LC-MS/MS: liquid chromatography–tandem mass spectrometry.

**Table 4 ijms-24-04109-t004:** Examples of metabolomic changes induced by AuNPs in cells.

Shape	Size	Surface Chemistry	Cell Type	Exposure	Method	Ref
Spheres	18 nm	citrate, poly-(sodium styrene sulfonate (PSSNa), or poly-vinylpyrrolidone (PVP)	HepG2 (human hepatocellular carcinoma)	0.25 nM (PSSNa and PVP) and 0.5 nM (citrate); 3 h	GC-MS ^1^ and LC-MS/MS ^2^	[89]
Spheres	20 nm	citrate	HDF (human normal dermal fibroblasts)	200 μM; 4, 8 and 24 h	LC-QTOF-MS ^3^	[90]
Rods	10 nm width, 40 nm length	Cetyltrimethylammonium Bromide (CTAB)	TM-4 (mouse Sertoli cells)	10 nm; 24 h	GC-MS ^1^	[92]
Spheres	50 nm	double-stranded DNA	PC3 and DU145 (human prostate cancer cell lines)	20 μg/mL; 24 h	LC-QTOF-MS ^3^	[93]

^1^ GC-MS: gas chromatography–mass spectrometry. ^2^ LC-MS/MS: liquid chromatography–tandem mass spectrometry. ^3^ LC-QTOF-MS: liquid chromatography coupled to a quadrupole-time-of-flight mass spectrometer.

**Table 6 ijms-24-04109-t006:** Examples of multi-omics studies showing changes induced by AuNPs in cells.

Shape	Size	Surface Chemistry	Cell Type	Exposure	Method	Ref
Spheres	5 nm; 30 nm	citrate	Caco 2 (human epithelial colorectal adenocarcinoma)	300 μM; 72 h	2D-DIGE/MS ^1^; LC-HRMS/MS ^2^	[137]
Spheres	5 nm; 20 nm	alkyl ammonium bromide, alkyl sodium carboxylate, or poly(ethylene glycol) (PEG)	THP-1 (human leukemia monocytic cell)	trancriptomics: 27 μg/mL (5 nm); 4 μg/mL (20 nm); 6 h. proteomics: 35 μg/mL (5 nm); 15 μg/mL (20 nm); 24 h	RNA-Seq; HPLC-MS ^3^	[138]

^1^ 2D-DIGE/MS: two-dimensional difference gel electrophoresis followed by mass spectroscopy. ^2^ LC-HRMS/MS: liquid chromatography high resolution–tandem mass spectrometry. ^3^ HPLC-MS: high-performance liquid chromatography flowed by mass spectroscopy.

## Data Availability

Not applicable.

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
