# Peer review of "Mechanistic Insights into the Biological Effects of Engineered Nanomaterials: A Focus on Gold Nanoparticles"

_ijms, 2023, doi:10.3390/ijms24044109_

Round 1

Reviewer 1 Report

I would like to congratulate the authors for conducting the present study. Here goes a few concerns:

The type of study should be mentioned in the title

The keywords should be placed in alphabetic order

Although the Reviews do not follow the traditional experimental design, they also need an aim and a rationale of a revision. Therefore I suggest the authors to place an aim and a proper rationale for the review in the end of the Introduction section.

In the Figure 1 legend I recommend to place also the author of the study that is being cited. The same applies to all the other Figure legends.

I suggest the authors to place a small section on Critical Analysis (a kind of summary) regarding the most relevant setbacks of the subject, just before the “4.Future Perspectives and Conclusions”

Despite these small issues, the overall review is fine, and covers a quite wide of issues related with the topic.

Reviewer 2 Report

The topic of the presented manuscript is obviously important. Understanding on detailed mechanisms of biological action of gold compounds / nanoparticles is urgently required. The introduction of the manuscript is relatively good, but the review itself is formalistic and disappointing. It just lists the recent advances in the field. The manuscript adds almost nothing to the facts that are already present in the cited literature. No specific mechanisms, which root in molecular science, are discussed; no analysis or comparison of the existing hypotheses is done.

Round 2

Reviewer 1 Report

Dear author. I have no more concerns. 

Author Response

We would like to thank Reviewer 1 for the time spent reading and reviewing the manuscript.

Reviewer 2 Report

The manuscript has been improved, but the major points have not been cleared. The main issue of the manuscript is the inconsistency between the very broad introduction and the quite specific main part, devoted to consequences of treatment of the cells with AuNPs. The main part is fairly well done technically, but it is not about “molecular mechanisms” of AuNP action (causes), but about “molecular consequences” of AuNP action (effects).

An example to emphasize the difference: suppose you treat eukaryotic cells with permissive dose of cyanide, and start to observe them by various omics approaches. It is almost certain, that you will observe multiple fast or deferred changes in metabolites, ROS production, mitochondrial failure, and, consequently, changes of gene expression, epigenomics, cell signaling, cell cycle regulation, apoptosis regulation, and other attempts of the cell to survive. Cyanide will be shown to react with hundreds of molecules in the cell. The response will probably depend on dose and cell type. But all this will hardly tell you about the true molecular mechanism, i.e. inhibition of cytochrome-c-oxidase by CN- binding in heme-a3–CuB center.

It does not mean that the molecular consequences are not important, but in my humble opinion it is not correct to call them "mechanistic understanding".

 Please consider the following points.

 1. The beginning Section 1 is very broad. It should be abridged and made more specific to AuNPs. "Nanomaterials in general" exceed the scope of this paper (especially if you consider viruses and liposomes as ENMs, why not consider genetically engineered proteins then, they are also nanoscale). Also, the success of nanomaterials is a well known fact by now. Actually, lines 27-116 are loosely connected to the specific topic of the manuscript (omic studies of AuNP effects). Lines 108-116 are completely unrelated. I suggest that the authors narrow the scope to AuNPs from the very beginning.

2. The title should be revised. It should indicate that the review is about gold nanoparticles (and cell cultures).

3. Line 24. “Molecular mechanisms” should be excluded from keywords. First, the paper is not about molecular mechanisms (see above); second – it is too broad. It can be replaced by smth. like “studies on cultured cells”.

4. Conclusion is also broad and non-specific. It should be more about AgNPs, than about some abstract nanomaterials. Some of the text from Lines 556-583 can be moved to conclusion.

Minor and technical points.

  • Figure 1 is poor for viewing. The font is very small, and it is impossible to read the text that is in the lower sectors, as it is curved and turned upside-down. Also, please check that the resolution in the final figures is high, now it is too low.
  • Line 53. The high precision of the dubious figure 2314.81 is puzzling.
  • Check if ref 6 is related to the manuscript topic.
  • Lines 139, 222 – are quotes really required? It is well known that inertness is a relative property
  • Line 155. Ref 37 mentions the history of ancient use of gold for treatment of diseases. It does not prove that the treatment was effective (further materials of section 2.1 hint that it was hardly effective)
  • Please double-check the tables for the correct use of “nm” (diameter, nanometers) and “nM”(concentration or dosage, nanomol/liter)
  • Line 278 – what is the mechanism?

It is also recommended that the authors added some comments on limitations of omics approach, i.e. curse of dimensionality. The challenge that lies in the proper interpretation of the biology or pathology represented by the high dimensional data should be mentioned.

On my personal experience with omic data, I would like to note that is produces more questions than answers if you don’t have a prior hypothesis. It is like obtaining an ounce of gold but in a thousand tons of slag. I wish the authors good luck in their omic studies.

Round 3

Reviewer 2 Report

The manuscript has been significantly improved and can be published. The text has become more focused and the title was amended.

 While I still think that the topic is not very novel, and the introduction is too broad (why not write about chemical properties of gold then, these are much less known now than the success of ENM), the authors have their point.

I also disagree that omic studies (correlation data) are mechanistic (cause-effect) studies, but I understood the authors’ point of view, especially as the paper is not so categorical about it now. Working at the edge of biochemistry and chemometrics, I know that nowadays many biologists have no respect for accurate terminology, and they are even proud of it.